# LAMP: LONG-TAILED MULTIMODAL PROMPT TUNING FOR VISION-LANGUAGE MODELS

## ABSTRACT

Prompt Learning (PL) has sparked growing interest as an efficient method for adapting large Vision-Language Models (VLMs) to downstream tasks. While most existing PL methods are designed for (nearly) balanced data, real-world datasets always exhibit a long-tailed distribution, which necessitates the design of PL methods specifically for imbalanced scenarios. This paper introduces our LAMP framework to enable VLMs better to learn from the long-tailed base classes and achieve non-biased predictions for both base and new classes. LAMP is integrated into specific intermediate layers of the frozen VLM, where for each layer we introduce three co-designed mechanisms: Multimodal Prompt Pool (MPP), Modality-Shared Prompts (MSP) and Load Balancing Optimization (LBO). MPP is motivated by the goal of boosting feature clustering to foster the mutual learning of head and tail classes, in which we introduce separated multimodal prompts that are dynamically combined into the model via similarity metrics. In addition to the prompts retrieved by MPP, we introduce globally shared MSPs to better adapt to cross-modal semantics and enhance the training robustness. Furthermore, to promote tight and discernible feature clustering via prompts, we treat each prompt as an expert and adopt a load-balancing technique from mixture-of-experts in LLMs, named LBO. LBO dynamically adjusts attention weights with an externally optimized bias, thereby making the activation of each prompt more evenly distributed and preventing the overfitting of head classes. Extensive experiments under various long-tailed settings demonstrate that our LAMP consistently outperforms other state-of-the-art methods. Code is available at Anonymous Github Link.

## 1 INTRODUCTION

Recent years have witnessed a striking advancement in Vision-Language Models (VLMs) (Radford et al., 2021; Jia et al., 2021; Li et al., 2022a), which has demonstrated unprecedented modeling and generation abilities by training over web-scale image-text corpora. As one of the pioneering VLMs, CLIP (Radford et al., 2021) is pre-trained over 400-million unlabeled image-text pairs with contrastive learning to establish a modality-joint feature space that captures cross-modal correspondences; Supported by the comprehensiveness and generality, it can be easily adapted to various downstream tasks, i.e., cross-modality generation (Ge et al., 2023; Chen et al., 2024; Zhang et al., 2021), image captioning (Barraco et al., 2022; Kornblith et al., 2023), retrieval (Baldrati et al., 2022; Zhang et al., 2024a), vision question answering (Keskar et al., 2025) and etc.

To efficiently adapt VLMs to specific datasets, an emerging trend is to conduct Prompt Learning (PL) (Lu et al., 2022; Zhou et al., 2022b;a; Zhang et al., 2024b; Shi et al., 2024b; Xu et al., 2025), which introduces several trainable parameters plugged into frozen VLMs to register task-specific knowledge. Early prompt learning methods, like CoOp (Zhou et al., 2022b), CoCoOp (Zhou et al., 2022a), learn isolated/conditioned textual prompt templates to guide task-aware modality alignment; MaPLe (Khattak et al., 2023a) extends this idea by incorporating layer-wise prompting, where textual prompts are employed for each layer and utilized to generate visual prompts. To avoid catastrophic forgetting caused by excessive prompt tuning, some methods (Zhu et al., 2023; Khattak et al., 2023b) specifically design self-supervision or gradient refinement techniques, where models are encouraged to narrow the discrepancy between the unbiased zero-shot CLIP. Recent methods, like MMA (Yang et al., 2024) and MMRL (Guo & Gu, 2025a;b), further enhance adaptation by explicitly modeling cross-modal interactions or integrating multiple tuning strategies.

Notably, most existing PL methods (Zhou et al., 2022b;a; Xu et al., 2023; Yang et al., 2024; Guo & Gu, 2025a;b) for VLMs are designed for (nearly) balanced multimodal datasets, which deviates from reality scenarios where data is universally long-tail distributed (Du et al., 2024b;a).This highlights that PL on VLMs should also be specifically tailored to these circumstances. Unfortunately, few methods have discussed this: some existing studies (Jia et al., 2022; Dong et al., 2022a; Xu et al.) propose visual prompt finetuning under long-tailed scenarios, but only for vision models; The method most closely related to our work is Candle (Shi et al., 2024a), which employs visual/textual prototypes as well as cross-modality attention to encourage balanced recognition of multimodal datasets. However, it only discusses the output-wise balancing techniques to facilitate the construction of prototypes for classification; *further insights on how refined prompts can help with the long-tailed problem, as well as utilizing modality interactions to achieve balanced learning, remain unexplored.*

**Contributions.** To alleviate the above issues, this paper introduces a novel Long-tAiled Multimodal Prompt learning (dubbed LAMP) approach. Our motivation stems from the observations of traditional long-tailed (LT) learning (Li et al., 2023; 2022b; Du et al., 2024a;b): despite differences in form, most methods aim to promote the formation of compact, cross-category feature clusters in the embedding space, thereby overcoming performance bias caused by class size. Following this underlying feature clustering philosophy, we naturally propose to use different prompts to cluster features in long-tailed

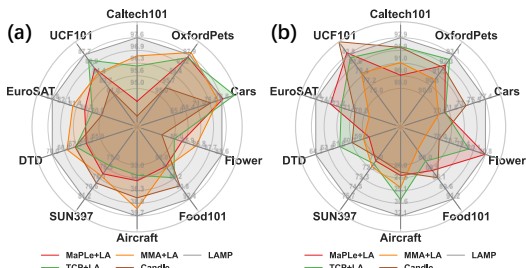

Figure 1: LAMP achieves significant improvements in accuracy (a) and macro-F1 (b) on long-tailed base-to-new benchmarks.

multimodal datasets, thereby boosting the mutual learning of head and tail classes during prompt finetuning. This leads to our design of the Multimodal Prompt Pool (MPP): MPP maintains several image-text prompt pairs; for each instance, we use its cls-token to retrieve the MPP and select the prompts based on the feature similarity. Furthermore, aware of the peculiarity of multimodal learning, we concatenate the retrieved prompts with another Modality-Shared Prompts (MSPs) to better capture cross-modal semantics. However, relying solely on the spontaneous feature clustering can still lead to overfitting of head-class features. Inspired by the load balancing in mixture-of-experts of Large Language Models, we further propose a Load Balancing Optimization (LBO) to equalize the activation levels across different prompts. The LBO technique is applied in the prompt querying stage; we first accumulate the attention rankings of each prompt across the dataset. Then, we introduce an externally optimized bias term, which is used to re-adjust the activation weights of prompts over the entire dataset. By optimizing this bias, we encourage the prompts to capture class-discriminative, compact, and evenly sized feature manifolds at the dataset level, thereby facilitating the long-tailed learning. Extensive experiments over 15 datasets under imbalanced settings demonstrate the consistent superiority of LAMP over existing methods (Figure 1, LAMP achieves significant improvements in accuracy and macro-F1). We summarize our contribution as follows:

- To the best of our knowledge, we first study the prompt learning (PL) of VLMs under long-tailed multimodal dataset settings, which yields more applicability and practical significance compared with existing PL methods.
- We propose the LAMP framework, incorporating three co-designed mechanisms to boost feature clustering and mutual learning of head and tail classes: Multimodal Prompt Pool (MPP), Modality-Shared Prompts (MSPs) and Load Balancing Optimization (LBO).
- Extensive experiments over 15 datasets under various imbalance settings show that LAMP consistently outperforms its competitors in accuracy and class rebalancing capabilities. Further ablation studies show that LAMP is efficient and the three mechanisms work synergistically.

## 2 RELATED WORKS

### 2.1 PROMPT LEARNING OF VLMS

The introduction of prompt learning on VLMs is built on its wide application across individual modalities (Jia et al., 2022; Li et al., 2024; Jiang et al., 2022; Dong et al., 2022b). Early methods

like CoOP (Zhou et al., 2022b) and CoCoOp (Zhou et al., 2022a) only add prompts in the text encoder. MaPLe (Khattak et al., 2023a) proposes to combine prompts in each layer of text encoders, and generate corresponding layer-wise image prompts from them; ProDA (Lu et al., 2022) and ProGrad (Zhu et al., 2023) discuss the supervision scheme of obtaining the optimal text prompts. ProVP (Xu et al., 2025) inserts layer-wise prompts into the forward flow in an incremental manner. DPL (Xu et al., 2023) decouples the attention within/between tokens and prompts, respectively, and finds that token-prompt cross-attention contributes most to the performance. DePT (Zhang et al., 2024b) raises a channel-adaptive classifier for prompt learning to mitigate the channel bias of domain generation. (Shi et al., 2024b) explores three correlation types between tokens and prompts when learning with missing modalities. The recent work MMRL (Guo & Gu, 2025a;b) proposes to insert shared prompts into high layers for fine-tuning, and further integrates multiple self-supervised methods as described above to achieve optimal performance. There are also theoretical studies, for example, (Wang et al., 2023; Hu et al., 2024) justify the theoretical limitations of prompt learning compared from the perspective of convex cones.

## 2.2 Long-tailed Learning

Real-world datasets follow a prevalent Long-Tailed (LT) distribution, where easy-to-acquire samples are always multiple times than rare ones. Typical LT solutions lie in reweighting (Menon et al., 2020; Cui et al., 2019; Cao et al., 2019; Ren et al., 2020; Tan et al., 2020), resampling (Kubat et al., 1997; Wallace et al., 2011; Chawla et al., 2002), contrastive learning (Zhu et al., 2022; Du et al., 2024b) and etc. For example, ProCo (Du et al., 2024b) borrows the mixture of von Mises-Fisher (vMF) distribution to model the long-tailed contrastive learning. Recently, some LLM-driven solutions have emerged; for example, LTGC (Zhao et al., 2024) leverages LLM-generated content to enrich tail representations; LPT (Dong et al., 2022a) introduces a group tuning technique over the vision models to alleviate the long-tailed effects. Candle (Shi et al., 2024a) explores the long-tailed multimodal dataset, and proposes an adapter tuning technique accompanied by logit-adjustment and class-prototypes. *In contrast to them, our study brings novel insights into further utilizing multimodal prompts over VLMs to achieve balanced learning of multimodal datasets.*

## 3 Methodology

### 3.1 Preliminaries

We first introduce the general settings of prompt learning on CLIP. Define the image and text branches of CLIP as $\mathcal{V}$ and $\mathcal{T}$ where both of them yield $N$ layers, i.e., $\mathcal{V} = \{V_i\}_{i=1}^N$ and $\mathcal{T} = \{T_i\}_{i=1}^N$; for a multimodal sample $\{x, y\}$ from the minibatch $\{\mathcal{X}, \mathcal{Y}\}$, we denote $B$ as the batchsize, $x \in \mathbb{R}^{H \times W \times 3}$ denotes the image, $t$ denotes the input text, $y$ denotes the label. We additionally use $\mathcal{C}$ to denote all classes. The image branch first divides $x$ into $N_v$ size-fixed patches, yielding patch embedding $E \in \mathbb{R}^{N_v \times D_v}$ ($D_v$ is the visual hidden dimension). Later, $E$ is jointly fed into cascaded encoder layers along with a fixed class embedding $cls$. The text branch first pads the input text $t$ with a predefined template (i.e., "a photo of a [class]") and tokenizes it, yielding $T \in \mathbb{R}^{N_t \times D_t}$ ($D_t$ is the textual dimension). Then, with the Begin-Of-Text (BOT) $bot$ and the End-Of-Text (EOT) $eot$ as indicators, $T$ is processed with cascaded $\{\mathcal{T}_i\}$. The process of each image and text encoder layer can be formalized as a recurrent form with the layer number omitted:

$$[E, cls] \leftarrow V([E, cls]); \quad [bot, T, eot] \leftarrow T([bot, T, eot]) \tag{1}$$

The visual and text output of the last layer, represented by $cls_N$, $eot_N$, are projected into a shared space with $P_v \in \mathbb{R}^{D_v \times D_s}$ and $P_t \in \mathbb{R}^{D_t \times D_s}$ ($D_s$ is the dimension of shared output space):

$$F = P_v(cls_N); \quad W = P_t(eot_N) \tag{2}$$

For the final classification stage, CLIP uses the final visual output $F$ and calculates its similarity with text outputs $\{W_c\}_{c=1}^C$ for each class to generate prediction $\boldsymbol{p}(y_c)$, i.e:

$$\boldsymbol{p}(y_c) = \sigma(\texttt{sim}(F, W_c)) = \frac{\exp(\texttt{sim}(F, W_c))}{\sum_{i=1}^C \exp(\texttt{sim}(F, W_i))} \tag{3}$$

Where $\sigma$ means the softmax. $\texttt{sim}(\cdot, \cdot)$ calculates the cosine similarity. For the training stage, $\boldsymbol{p}(y_c)$ is pushed to learn the label $y$ with the cross-entropy loss $\mathcal{L}_{CE}$.

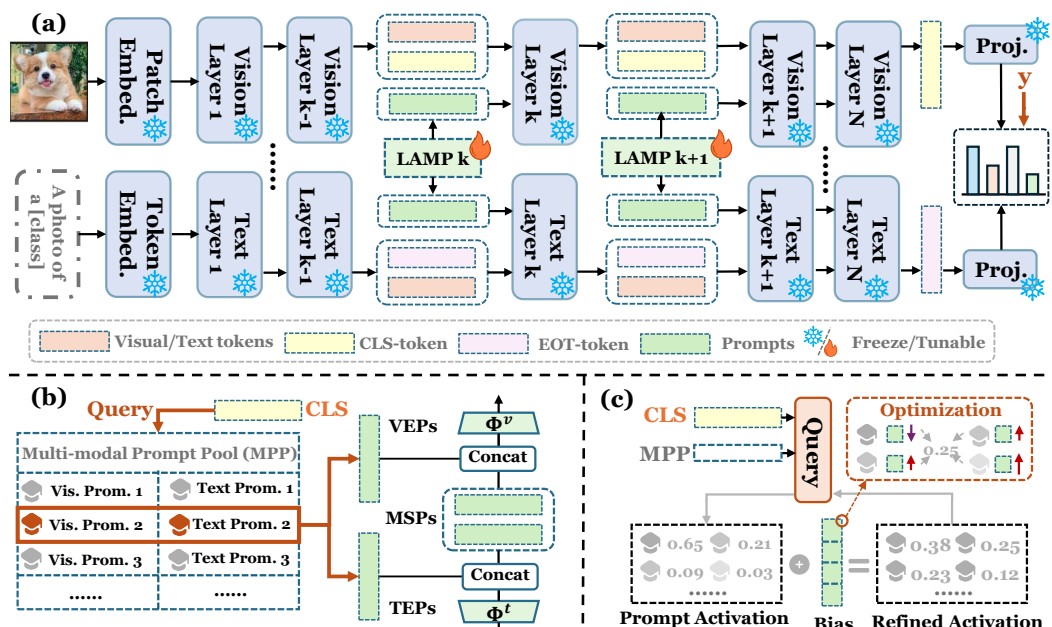

Figure 2: **(a)** The overview of LAMP, where we leverage prompts generated from LAMP modules in specific layers to dynamically balance the long-tailed effects. Each module yields three key designs: multimodal prompt pool, modality-shared prompts and load balancing optimization. **(b)** The LAMP module. where we query MPP with cls-token and concat the retrieved VEPs/TEPS with our MSPs. **(c)** Details of LBO, which is applied in the query operation. We optimize the extra bias to enforce even activations of different prompts.

## 3.2 LAMP: Long-tailed Multimodal Prompt learning

This section introduces our framework, consisting of cascaded LAMP modules plugged into the visual and text encoders. Each LAMP module comprises three key designs: *Multimodal Prompt Pool (MMP)*, *Modality-Shared Prompts (MSP)* and *Load Balancing Optimization (LBO)*.

**Multimodal Prompt Pool (MMP).** An underlying principle of traditional long-tail (LT) methods is to promote feature clustering between different samples and classes, thereby breaking the constraints of class size and enhancing the learning of both head and tail classes. Similarly, in our prompt learning scenario, we aim to use prompts to dynamically foster the formation of consistent and distinct clusters across the whole long-tailed dataset. A straightforward idea of leveraging prompts to boost feature clustering is to divide the dataset into multiple clusters based on the feature similarity, then employ distinct prompts for different clusters. Based on this motivation, we maintain a *Multimodal Prompt Pool (MMP)* $\mathcal{S}$ for each selected layer, containing $M$ prompt pairs $\{(\boldsymbol{v}_1, \boldsymbol{t}_1), (\boldsymbol{v}_2, \boldsymbol{t}_2), ..., (\boldsymbol{v}_M, \boldsymbol{t}_M)\}$ where $\boldsymbol{v} \in \mathbb{R}^D$ is the visual prompt, $\boldsymbol{v}_K \in \mathbb{R}^D$ is the text prompt. For each input $\boldsymbol{x}$, we obtain its cls-token $cls$ in the current layer (we don't specify layer notation $i$ for ease of representation), then leverage this token to query $K$ prompts from the prompt pool $\mathcal{S}$:

$$\texttt{Ret} = \{(\boldsymbol{v}_m, \boldsymbol{t}_m); \ k \in \texttt{ArgTop}(\texttt{sim}(cls_i, \{\boldsymbol{v}_m\}_{m=1}^M); K)\} \quad (4)$$

Where $\texttt{ArgTop}(x; K)$ calculates the index of elements that are within the top-$K$ entries of $x$. We concat these retrieved prompts $\texttt{Ret}$ separately by visual and textual splits, generating our Visual Extra Prompts (VEPs) $\boldsymbol{v}_{\text{ex}}$ and Textual Extra Prompts (TEPs) $\boldsymbol{t}_{\text{ex}}$:

$$\boldsymbol{v}_{\text{ex}} = \texttt{Concat}(\{\boldsymbol{v}_m; \boldsymbol{v}_m \in \texttt{Ret}\}); \ \boldsymbol{t}_{\text{ex}} = \texttt{Concat}(\{\boldsymbol{t}_m; \boldsymbol{t}_m \in \texttt{Ret}\}) \quad (5)$$

In the next part, we introduce another prompt that synergistically functions with VEPs and TEPs to better adapt the cross-modal knowledge.

**Modality-Shared Prompts (MSP).** Inspired by previous studies (Khattak et al., 2023a; Guo & Gu, 2025a) showing that cross-modality prompts exert better capability in capturing multimodal

semantics, we here introduce additional *Modality-Shared Prompts (MSP)* in addition to further enhance the model's perception of cross-modal semantic information. Denote $\boldsymbol{v}_{\text{sh}} \in \mathbb{R}^{S \times D}$ as MSP where $S$ represents its length. Then, $\boldsymbol{v}_{\text{sh}}$ is concatenated separately with $\boldsymbol{v}_{\text{ex}}, \boldsymbol{t}_{\text{ex}}$, then processed with separate projector $\boldsymbol{\phi}_v$ and $\boldsymbol{\phi}_t$, yielding results $P^v, P^t$ :

$$P^v = \phi^v(\text{Concat}(\boldsymbol{v}_{\text{ex}}, \boldsymbol{s})); \quad P^t = \phi^t(\text{Concat}(\boldsymbol{t}_{\text{ex}}, \boldsymbol{s})); \tag{6}$$

$P_v, P_t$ are then inserted into the forward process of the current layer. We can reformulate Eq. 1 as:

$$[E, P^v, cls] \leftarrow V([E, P^v, cls]); \quad [bot, P^t, T, eot] \leftarrow T([bot, P^t, T, eot]) \tag{7}$$

In the next chapter, we introduce another prompt that synergistically functions with VEPs and TEPs to better adapt the cross-modal knowledge.

**Load Balancing Optimization.** In the previous chapters, we promote feature re-clustering and interaction by dynamically assigning different prompts to features. However, leveraging models to spontaneously learn feature clustering from the datasets still tends to assign tighter manifolds to head-class features. To further enhance the distinction of clusters, inspired by the well-known load balancing (Wang et al., 2024; Qiu et al., 2025) mechanism in Mixture-of-Experts (MoEs) of Large Language Models (LLMs), we similarly propose a **Load Balancing Optimization (LBO)**: specifically, we treat each prompt in the pool as an expert, and the accumulated probability of each expert being selected is regarded as the expert activation values. Our goal is to encourage a consistent and uniform distribution of expert activations across all samples, thereby improving the balance and separation of features preserved by each expert. In our implementation, we modify the querying operation in Eq. 4 with an extra bias vector $\boldsymbol{b} \in \mathbb{R}^{M \times D}$ (Recall $M$ is the prompt number in MPP):

$$\text{Ret} = \{(\boldsymbol{v}_m, \boldsymbol{t}_m); \ m \in \text{ArgTop}(\text{sim}(cls_i, \{\boldsymbol{v}_m\}_{m=1}^M) + \boldsymbol{b}; K)\} \tag{8}$$

We rely on $\boldsymbol{b}$ to explicitly adjust the load of each expert (i.e, prompt). Notably $\boldsymbol{b}$ does not participate in the forward pass or gradient computation, but updated by an additional process at each step $t$:

$$\boldsymbol{b}_i \leftarrow \boldsymbol{b}_i + \gamma \cdot \text{sign}(\overline{\boldsymbol{a}}_{i,t}(K) - \boldsymbol{a}_{\text{cons}}(K)) \tag{9}$$

Recall $K$ is the retrieved prompt number from the prompt pool. $\text{sign}$ is the sign function, $\gamma$ is the learning rate; as indicated by (Wang et al., 2024), we employ an cosine scheduler to adjust $\gamma$, thereby reducing its variation when the model's update converges. In the subsequent ablation studies, we only investigate the impact of $\gamma$'s initial value. $\overline{\boldsymbol{a}}_{i,t}(K)$ and $\boldsymbol{a}(K)$ are core load metrics: $\overline{\boldsymbol{a}}_{i,t}(K)$ represents the activations of expert $i$ as we described above, which is iteratively updated with the number of times expert $i$ was among the top-$K$ items in the attention score up to the iteration $t$:

$$\overline{\boldsymbol{a}}_{i,t+1}(K) = \frac{G-1}{G}\overline{\boldsymbol{a}}_{i,t}(K) + \frac{1}{G}\sum_{\boldsymbol{x} \in \mathcal{X}} \mathbb{1}(m \in \text{ArgTop}(\text{sim}(cls_i, \{\boldsymbol{v}_{m'}\}_{m'=1}^M); K)) \tag{10}$$

Where $G$ counts the update time, $\mathbb{1}(x)$ is the indicator function; when $x$ is true, it's 1; else 0. This update mechanism allows us to comprehensively track the load distribution across the entire dataset. Additionally, the counter is reinitialized to zero upon the beginning of each epoch. Focusing on the $\boldsymbol{a}_{\text{cons}}(K)$ in Eq. 9, it's a constant and represents the average load conditioned on $M$. In our case, it simply equals $K/M$. By subtracting $\boldsymbol{a}_{\text{cons}}(K)$ in Eq. 9, we treat it as a baseline and adjust the load of all experts to be evenly distributed. The next chapter will summarize the objectives of LAMP.

**Learning Objectives.** In this part, we summarize the training process of LAMP. For the prompts retrieved from MPP, we maximize their similarity with the query to encourage the formation of compact and well-separated feature clusters in the dataset. This loss function is combined with the CE function as follows:

$$\mathcal{L}_{\text{LAMP}} = \mathcal{L}_{\text{CE}}(\boldsymbol{x}, y) + \beta \sum_{i \in \text{Sec}} (1 - \frac{1}{K} \sum_{\boldsymbol{v}_k \in \text{Ret}} \text{sim}(cls_i, \boldsymbol{v}_k)) \tag{11}$$

Where $\beta$ is a hyper-parameter for controlling the loss contributions; Sec is the selected layer for adding our LAMP module. Additionally, we further employ the LBO introduced above to balance the load distribution of different prompts. Considering that most PL tasks follow a base-to-new evaluation benchmark, i.e., training on labeled base classes and unlabeled data from new classes, followed by testing on base and new test sets separately, we further adjust $\boldsymbol{b}$ specifically for the new classes. Concretely, after training on the base classes, we directly perform inference on the unlabeled training set of new classes, during which we adjust $\boldsymbol{b}$ to equalize the load of different expert prompts on the new class data.

Table 1: **Results on long-tailed B2N benchmarks with an imbalance ratio of 10,20,50 respectively averaged over 5 runs**. We report the harmonic mean of accuracy (HM-A) and harmonic mean of macro-F1 (HM-M) on base and new classes. The best results are marked with **bold**.

| Method \| imb = 10 | Caltech101 | | Oxfordflowers | | Stanfordcars | | Flower102 | | Food101 | | FGVC-Aircraft | |
|---|---|---|---|---|---|---|---|---|---|---|---|---|
| | HM-A | HM-M | HM-A | HM-M | HM-A | HM-M | HM-A | HM-M | HM-A | HM-M | HM-A | HM-M |
| CoOp+LA | 91.78 | 90.96 | 94.10 | 90.44 | 69.23 | 66.44 | 71.86 | 64.42 | 89.25 | 88.83 | 31.12 | 23.41 |
| CoCoOp+LA | 95.09 | 91.23 | 96.69 | 91.91 | 71.71 | 67.19 | 77.61 | 73.68 | 91.20 | 89.02 | 33.71 | 26.15 |
| LFA | 95.69 | 90.72 | 94.09 | 89.60 | 72.72 | 68.72 | 84.38 | 78.16 | 90.44 | 88.64 | 34.27 | 27.76 |
| MaPLe+LA | 95.00 | 90.60 | 97.64 | 92.46 | 76.65 | 72.40 | 86.24 | 80.80 | 88.36 | 87.74 | 35.98 | 28.22 |
| TCP+LA | 96.43 | 92.31 | 97.60 | 92.31 | **78.80** | 75.01 | 85.78 | 78.66 | 89.79 | 88.60 | 35.41 | 27.90 |
| MMA+LA | 96.84 | 92.36 | **97.88** | 92.80 | 75.75 | 72.48 | 87.09 | 80.04 | 88.49 | 86.63 | 38.93 | 30.56 |
| Candle | 94.40 | 91.37 | 95.99 | 91.59 | 74.30 | 71.69 | 85.03 | 78.05 | 90.80 | 85.29 | 37.78 | 29.40 |
| LAMP | **97.58** | **92.91** | 97.76 | **93.09** | 76.45 | **77.37** | **88.65** | **80.81** | **92.45** | **91.15** | **40.69** | **32.13** |

| Method \| imb = 10 | SUN397 | | DTD | | EuroSAT | | UCF101 | | ImageNet | | ✦ Average ✦ | |
|---|---|---|---|---|---|---|---|---|---|---|---|---|
| | HM-A | HM-M | HM-A | HM-M | HM-A | HM-M | HM-A | HM-M | HM-A | HM-M | HM-A | HM-M |
| CoOp+LA | 72.15 | 68.38 | 54.88 | 40.97 | 54.42 | 49.84 | 64.74 | 63.32 | 68.05 | 62.45 | 69.23 | 64.50 |
| CoCoOp+LA | 77.99 | 70.98 | 65.11 | 56.35 | 60.28 | 54.48 | 76.78 | 72.91 | 70.19 | 64.38 | 74.21 | 68.93 |
| LFA | 78.39 | 72.44 | 67.43 | 60.77 | 69.56 | 57.79 | 82.71 | 78.42 | 70.44 | 64.80 | 76.37 | 70.71 |
| MaPLe+LA | 78.16 | 72.21 | 67.20 | 59.60 | 81.29 | 74.80 | 85.90 | 82.27 | 71.78 | 65.30 | 78.56 | 73.31 |
| TCP+LA | 77.15 | 71.92 | 68.07 | 61.03 | 80.94 | 72.24 | 87.00 | 83.79 | 70.24 | 64.19 | 78.84 | 73.45 |
| MMA+LA | 78.43 | 72.46 | 68.80 | 62.07 | 81.84 | 73.68 | 85.30 | 81.61 | 69.15 | 63.77 | 78.95 | 73.50 |
| Candle | 79.26 | 72.70 | 68.13 | 60.07 | 80.51 | 70.72 | 83.17 | 79.52 | 68.89 | 62.10 | 78.02 | 72.05 |
| LAMP | **81.18** | **75.66** | **70.02** | **64.10** | **84.78** | **76.36** | **87.70** | 82.02 | **72.96** | **66.67** | **80.93** | **75.66** |

| Method \| imb = 20 | Caltech101 | | Oxfordflowers | | Stanfordcars | | Flower102 | | Food101 | | FGVC-Aircraft | |
|---|---|---|---|---|---|---|---|---|---|---|---|---|
| | HM-A | HM-M | HM-A | HM-M | HM-A | HM-M | HM-A | HM-M | HM-A | HM-M | HM-A | HM-M |
| CoOp+LA | 92.65 | 90.92 | 94.15 | 90.68 | 67.39 | 62.31 | 73.72 | 67.61 | 86.38 | 86.08 | 29.33 | 23.16 |
| CoCoOp+LA | 95.25 | 92.49 | 96.64 | 92.08 | 71.38 | 70.32 | 80.31 | 72.42 | 91.20 | 88.42 | 32.78 | 26.44 |
| LFA | 95.56 | 92.12 | 90.90 | 86.45 | 70.35 | 68.36 | 84.03 | 78.51 | 89.72 | 85.92 | 33.02 | 27.05 |
| MaPLe+LA | 96.09 | 92.51 | 96.29 | 92.19 | 74.88 | 71.20 | 87.84 | 81.10 | 90.54 | 87.98 | 34.88 | 28.07 |
| TCP+LA | 97.78 | 93.76 | 95.75 | 92.01 | 73.34 | 70.45 | 87.27 | 80.98 | 89.90 | 86.40 | 36.26 | 30.31 |
| MMA+LA | 97.60 | 93.57 | 96.81 | 92.20 | **74.54** | 71.79 | 86.85 | 79.57 | 91.00 | 88.16 | 34.40 | 28.47 |
| Candle | 95.84 | 92.46 | 95.89 | 91.77 | 73.49 | 69.70 | 84.92 | 76.26 | 90.75 | 87.51 | 38.02 | 32.63 |
| LAMP | **97.88** | **93.92** | **96.87** | **93.56** | 74.46 | 71.99 | **88.95** | **82.89** | **91.62** | **89.02** | **39.04** | **34.54** |

| Method \| imb = 20 | SUN397 | | DTD | | EuroSAT | | UCF101 | | ImageNet | | ✦ Average ✦ | |
|---|---|---|---|---|---|---|---|---|---|---|---|---|
| | HM-A | HM-M | HM-A | HM-M | HM-A | HM-M | HM-A | HM-M | HM-A | HM-M | HM-A | HM-M |
| CoOp+LA | 68.93 | 67.38 | 55.18 | 38.66 | 62.64 | 51.01 | 60.12 | 57.60 | 67.01 | 60.05 | 68.86 | 63.22 |
| CoCoOp+LA | 77.29 | 69.03 | 61.31 | 52.16 | 58.82 | 42.93 | 71.70 | 68.36 | 69.58 | 63.50 | 73.30 | 67.10 |
| LFA | 77.30 | 69.45 | 66.07 | 56.51 | 68.74 | 55.22 | 81.80 | 79.51 | 69.20 | 63.11 | 75.15 | 69.29 |
| MaPLe+LA | 78.80 | 70.02 | 67.47 | 57.50 | 83.28 | 73.42 | 83.25 | **82.60** | 69.95 | 63.80 | 78.48 | 72.76 |
| TCP+LA | 78.23 | 69.95 | 67.14 | 56.97 | 84.51 | 72.35 | 81.91 | 79.64 | 68.87 | 62.58 | 78.27 | 72.31 |
| MMA+LA | 76.75 | 67.28 | 67.64 | 58.22 | 83.79 | 73.27 | 80.76 | 78.28 | 68.03 | 61.97 | 78.02 | 72.07 |
| Candle | 78.53 | 70.11 | 67.32 | 57.23 | 80.96 | 71.01 | 82.59 | 81.51 | 67.50 | 60.80 | 77.80 | 71.91 |
| LAMP | **79.59** | **71.23** | **69.30** | **60.01** | **84.65** | **75.40** | **83.46** | 82.23 | **71.20** | **65.49** | **79.73** | **74.57** |

| Method \| imb = 50 | Caltech101 | | Oxfordflowers | | Stanfordcars | | Flower102 | | Food101 | | FGVC-Aircraft | |
|---|---|---|---|---|---|---|---|---|---|---|---|---|
| | HM-A | HM-M | HM-A | HM-M | HM-A | HM-M | HM-A | HM-M | HM-A | HM-M | HM-A | HM-M |
| CoOp+LA | 93.30 | 89.46 | 93.34 | 89.09 | 67.18 | 61.48 | 75.45 | 67.73 | 87.65 | 84.20 | 29.20 | 20.17 |
| CoCoOp+LA | 94.90 | 89.70 | 95.44 | 91.02 | 69.97 | 64.32 | 76.84 | 68.67 | 91.10 | 89.53 | 31.45 | 22.09 |
| LFA | 94.23 | 88.09 | 86.76 | 85.51 | 67.95 | 62.17 | 82.81 | 72.80 | 87.73 | 84.65 | 30.75 | 21.25 |
| MaPLe+LA | 95.57 | 90.70 | **96.85** | 91.80 | 71.96 | 66.60 | 85.06 | 78.10 | 89.70 | 86.05 | 33.40 | 25.89 |
| TCP+LA | 95.39 | 90.49 | 95.96 | 90.56 | 70.81 | 65.35 | 86.67 | 78.39 | 90.03 | 87.17 | 32.72 | 24.32 |
| MMA+LA | 96.43 | 91.23 | 96.89 | 91.85 | 72.48 | 66.84 | 85.79 | 77.40 | 89.05 | 85.63 | 35.98 | 27.21 |
| Candle | 94.95 | 90.40 | 95.83 | 91.33 | 71.78 | 66.30 | 84.62 | 76.03 | 90.70 | 87.76 | 36.68 | 28.13 |
| LAMP | **97.09** | **92.54** | 96.35 | **92.85** | **73.79** | **68.41** | **88.25** | **80.33** | **91.20** | **88.87** | **38.40** | **30.20** |

| Method \| imb = 50 | SUN397 | | DTD | | EuroSAT | | UCF101 | | ImageNet | | ✦ Average ✦ | |
|---|---|---|---|---|---|---|---|---|---|---|---|---|
| | HM-A | HM-M | HM-A | HM-M | HM-A | HM-M | HM-A | HM-M | HM-A | HM-M | HM-A | HM-M |
| CoOp+LA | 65.91 | 60.78 | 51.42 | 36.11 | 57.35 | 50.29 | 61.62 | 53.66 | 65.59 | 59.59 | 68.00 | 61.14 |
| CoCoOp+LA | 76.18 | 69.97 | 59.37 | 44.76 | 64.99 | 58.91 | 77.53 | 68.53 | 68.70 | 61.34 | 73.32 | 66.26 |
| LFA | 75.13 | 69.05 | 61.78 | 47.99 | 61.91 | 55.15 | 79.49 | 71.79 | 68.21 | 61.01 | 72.43 | 65.41 |
| MaPLe+LA | 76.80 | 70.10 | 63.40 | 52.90 | 78.60 | 68.03 | 79.30 | 72.40 | 69.36 | 62.10 | 76.36 | 69.52 |
| TCP+LA | 77.20 | 69.44 | 66.78 | 56.95 | 81.87 | 71.09 | 79.93 | 71.09 | 68.05 | 60.79 | 76.86 | 69.69 |
| MMA+LA | 77.91 | 68.35 | 66.10 | 56.18 | **83.41** | 74.37 | 80.86 | 73.25 | 67.20 | 59.25 | 77.46 | 70.14 |
| Candle | 78.05 | 69.29 | 65.69 | 55.37 | 80.17 | 70.89 | 81.72 | 74.00 | 66.75 | 58.40 | 76.99 | 69.81 |
| LAMP | **78.50** | **70.22** | **66.83** | **59.35** | 83.28 | 74.10 | **82.50** | **78.78** | **70.45** | **64.67** | **78.79** | **72.76** |

# 4 EXPERIMENTS

## 4.1 SETTINGS

**Benchmarks.** Following previous approaches, we employ four well-known benchmarks but manipulate them to follow the long-tailed settings: ❶ Base-to-New (B2N) benchmark, which separates the whole dataset into disjoint base to new classes. Models are trained with labeled base-class training sets, and evaluated on test sets of both base and new classes; for this dataset, we employ 2 object recognition datasets ImageNet (Deng et al., 2009) and Caltech101 (Fei-Fei et al., 2004); 1 scene understanding dataset SUN397 (Xiao et al., 2010); 1 texture dataset DTD (Cimpoi et al., 2014); 1 satellite dataset EuroSAT (Helber et al., 2019), 1 action dataset UCF101 (Soomro et al., 2012) and 5 fine-grained recognition datasets: OxfordPets (Parkhi et al., 2012), StanfordCars (Krause et al., 2013), Flowers102 (Nilsback & Zisserman, 2008), Food101 (Bossard et al., 2014) and FGVC-Aircraft (Maji et al., 2013). The training data is manipulated into long-tailed distribution by down-sampling the base

Table 2: **Results on few-shot benchmark averaged over 5 runs.** We report the accuracy of base (Base), new classes (New) and their harmonic mean (HM). The best results are marked with **bold**.

| Method | ✦ Average ✦ | | | ImageNet | | | Caltech101 | | | OxfordPets | | | StanfordCars | | | Flowers102 | | |
|---|---|---|---|---|---|---|---|---|---|---|---|---|---|---|---|---|---|---|
| | Base | New | HM | Base | New | HM | Base | New | HM | Base | New | HM | Base | New | HM | Base | New | HM |
| CoOp | 82.69 | 63.22 | 70.83 | 76.47 | 67.88 | 71.92 | 98.00 | 89.81 | 93.73 | 93.67 | 95.29 | 94.47 | 78.12 | 60.40 | 68.13 | 97.60 | 59.67 | 74.06 |
| CoCoOp | 80.47 | 71.69 | 75.44 | 75.98 | 70.43 | 73.10 | 97.96 | 93.81 | 95.84 | 95.20 | 97.69 | 96.43 | 70.49 | 73.59 | 72.01 | 94.87 | 71.75 | 81.71 |
| MaPLe | 82.28 | 75.14 | 78.24 | 76.66 | 70.54 | 73.47 | 97.74 | 94.36 | 96.02 | 95.43 | 97.76 | 96.18 | 72.94 | 74.00 | 73.47 | 95.92 | 72.46 | 82.56 |
| TCP | 83.20 | 76.94 | 79.72 | 77.31 | 71.00 | 74.02 | **98.40** | 94.00 | 96.15 | 95.40 | **98.07** | **96.72** | 78.50 | 73.10 | 75.70 | 97.77 | 75.93 | 85.48 |
| MMA | 84.13 | 75.36 | 79.26 | 77.27 | 69.87 | 73.38 | 98.23 | 94.67 | **96.42** | 94.67 | 97.20 | 95.92 | 80.80 | 74.13 | 77.32 | 97.73 | 75.57 | 85.23 |
| Candle | 83.88 | 76.53 | 79.87 | 76.97 | 68.54 | 72.48 | 98.54 | 94.47 | 96.46 | 95.53 | 97.34 | 96.43 | 79.14 | 74.92 | 76.97 | 98.01 | 77.52 | 86.57 |
| LAMP | **85.23** | **77.18** | **80.83** | **77.62** | **71.60** | **74.49** | 98.08 | **94.02** | 96.01 | **95.65** | 97.82 | 96.72 | **81.09** | **75.40** | **78.14** | **98.04** | **77.98** | **86.87** |

| Method | Food101 | | | FGVCAircraft | | | SUN397 | | | DTD | | | EuroSAT | | | UCF101 | | |
|---|---|---|---|---|---|---|---|---|---|---|---|---|---|---|---|---|---|---|
| | Base | New | HM | Base | New | HM | Base | New | HM | Base | New | HM | Base | New | HM | Base | New | HM |
| CoOp | 88.33 | 82.26 | 85.19 | 40.44 | 22.30 | 28.75 | 80.60 | 65.89 | 72.51 | 79.44 | 41.18 | 54.24 | 92.19 | 54.74 | 68.69 | 84.69 | 56.05 | 67.45 |
| CoCoOp | 90.70 | 91.29 | 90.99 | 33.41 | 23.71 | 27.74 | 79.74 | 76.86 | 78.27 | 77.01 | 56.00 | 64.85 | 87.49 | 60.04 | 71.21 | 82.33 | 73.45 | 77.64 |
| MaPLE | 90.71 | 92.05 | **91.38** | 37.44 | 35.61 | 36.50 | 80.82 | 78.70 | 79.75 | 80.36 | 59.18 | 68.16 | 94.07 | 73.23 | 82.35 | 83.00 | 78.66 | 80.77 |
| MMA | 90.13 | 91.30 | 90.71 | 40.57 | 36.33 | 38.33 | 82.27 | 78.57 | 80.38 | 83.20 | **65.63** | 73.38 | 85.46 | **82.34** | 83.87 | 86.23 | 80.03 | 83.01 |
| TCP | **90.57** | **91.37** | 90.97 | 41.97 | 34.43 | 37.83 | 82.63 | 78.20 | 80.35 | 82.77 | 58.07 | 68.25 | 91.63 | 74.73 | 82.32 | 87.13 | **80.77** | 83.83 |
| Candle | 90.52 | 91.23 | 90.87 | 43.86 | **36.69** | 39.96 | 81.64 | 77.93 | 79.74 | 81.40 | 61.35 | 69.97 | 89.97 | 81.33 | 85.43 | 87.13 | 80.51 | 83.69 |
| LAMP | 90.40 | 91.03 | 90.71 | **45.95** | 36.66 | **40.78** | **82.92** | **79.00** | **80.91** | **85.04** | 65.26 | **73.85** | **93.90** | 80.42 | **86.63** | **88.79** | 79.83 | **84.07** |

classes. ❷ Few-Shot Benchmark, which assesses the model's generalization capability under few-shot (16-shot in default) settings. This benchmark employ the same dataset as B2N benchmark, but without the long-tailed manipulation; ❸ Long-tailed Cross-Dataset Transfer Benchmark, which assesses the models' generalization capability across datasets, The models are first trained on long-tailed ImageNet, then evaluated on other 10 datasets above; ❹ Long-tailed Domain Generalization Benchmark, which assesses the models' generalization capability across different domains. The models are first trained on long-tailed ImageNet, then evaluated on four components of ImageNet, i.e., ImageNet-A (Hendrycks et al., 2021b), ImageNet-S (Wang et al., 2019), ImageNet-R (Hendrycks et al., 2021a) and ImageNetV2 (Recht et al., 2019).

As for the concrete steps for applying long-tailed distributions to the dataset, we conduct down-sampling to manipulate the size of all classes into an exponential decayed long-tail curve (Shi et al., 2024a) satisfying a certain imbalance ratio imb (if the sample number of class $i$ is $|\mathcal{C}_i|$, the imbalance ratio is calculated with $\text{imb} = \max(|\mathcal{C}_i|)/\min(|\mathcal{C}_i|)$). In down-sampling, we fix the $max(|\mathcal{C}_i|)$ as the size of the maximum class in the original data and apply different imbalance ratios. By default, we treat the top 20% classes and others as tail classes.

▶ **Implementation Details.** We use a pretrained CLIP equipped with ViT-B/16 for all datasets. The batch size for training and inference are 128 and 100, respectively. Models are finetuned for 10 to 100 epochs (depending on the dataset) with an Adam optimizer equipped with a unified 3e-4 learning rate, 5e-4 weight decay, and 0.9 momentum. The length and depth of shared prompts are set to 9(3→12) and 2, respectively. The length for extra visual and textual prompts is 2; $\beta$=1; All prompt dimension is set to 32, and we have a concrete discussion on this in the appendix. Baselines are equipped with their provided settings. We compare LAMP with typical prompt learning methods like CoOp (Zhou et al., 2022b), CoCoOp (Zhou et al., 2022a), LFA and MaPLe (Khattak et al., 2023a); We also consider several recent state-of-arts like Candle (Shi et al., 2024a), TCP (Yao et al., 2024) and MMA (Yang et al., 2024). Notably, since these methods aren't crafted for long-tailed circumstances expect for Candle, we follow (Shi et al., 2024a) to adopt a Logit-Adjusted Loss (Menon et al., 2020) to ensure fairness. LFA is only compared in the first two benchmarks due to its limited compatibility with other settings. More details are in **Appx. C**.

## 4.2 EXPERIMENT RESULTS

▶ **Results of long-tailed B2N benchmark.** We report the comparison results in Table 1. Accordingly, we can conclude that LAMP generally yields the best accuracy and macro-F1 compared with other methods with different imbalance ratios. For example, focusing on the previous state-of-the-arts MMA+LA and Candle, LAMP outperforms them by a maximal margin of 2.91%/3.61% and 1.97%/2.16% in accuracy/macro-F1 respectively. Also, we find that the performance gap between text-only methods CoOp / CoCoOp and others is narrowed with the increment of the imbalance ratio, and the performance degradation of MaPLe, and MMA is especially considerable. This highlights that text information is crucial for keeping the balanced performance on long-tailed datasets. We also observe that LAMP yields a greater performance gain in Macro-F1 than in accuracy, particularly as the imbalance setting worsens. This indicates that our model is more adept at balancing the acquired knowledge under extreme conditions than other methods.

Table 3: **Results of Long-tailed Cross-dataset transfer benchmark** averaged over 5 runs. All models are trained on long-tailed ImageNet with `imb = 100`, then evaluated over another target dataset. We report the accuracy and macro-F1. The best results are marked with **bold**.

| Method | ImageNet (Source) | | Caltech101 | | Oxfordflowers | | Stanfordcars | | Flower102 | | Food101 | |
|---|---|---|---|---|---|---|---|---|---|---|---|---|
| | Acc. | Mac. | Acc. | Mac. | Acc. | Mac. | Acc. | Mac. | Acc. | Mac. | Acc. | Mac. |
| CoOp+LA | 70.75 | 63.92 | 90.85 | 88.78 | 86.85 | 82.93 | 64.95 | 61.67 | 67.12 | 59.41 | 85.02 | 83.85 |
| CoCoOp+LA | 71.24 | 64.40 | 91.31 | 89.39 | 88.76 | 84.14 | 65.52 | 63.00 | 69.50 | 61.50 | 86.30 | 84.12 |
| MaPLe+LA | 70.18 | 63.35 | 90.17 | 88.02 | 89.29 | 85.49 | 65.09 | 62.43 | 70.38 | 61.87 | 86.05 | 84.00 |
| TCP+LA | 71.03 | 64.34 | 90.56 | 88.10 | 89.37 | 85.66 | 64.78 | 62.36 | 71.15 | 62.26 | 86.45 | 84.17 |
| MMA+LA | 70.65 | 63.98 | 90.21 | 87.51 | 88.29 | 84.38 | 63.87 | 62.01 | 71.10 | 62.23 | 85.80 | 83.81 |
| Candle | 70.92 | 64.62 | 91.40 | 89.78 | 88.96 | 85.82 | 64.60 | 62.57 | 68.94 | 58.51 | 85.39 | 83.25 |
| LAMP | **72.70** | **66.28** | **91.85** | **90.14** | **90.12** | **87.07** | 65.48 | 62.98 | **72.26** | **63.40** | **86.58** | **84.31** |

| Method | FGVC-Aircraft | | SUN397 | | DTD | | EuroSAT | | UCF101 | | Avg. (w/o Source) | |
|---|---|---|---|---|---|---|---|---|---|---|---|---|
| | Acc. | Mac. | Acc. | Mac. | Acc. | Mac. | Acc. | Mac. | Acc. | Mac. | Acc. | Mac. |
| CoOp+LA | 18.92 | 12.00 | 63.25 | 59.92 | 42.46 | 32.71 | 44.40 | 40.69 | 65.98 | 64.70 | 63.69 | 59.14 |
| CoCoOp+LA | 22.80 | 14.82 | 66.07 | 61.73 | 45.04 | 35.64 | 42.86 | 38.34 | 67.54 | 66.45 | 65.18 | 60.32 |
| MaPLe+LA | 23.95 | 14.49 | 66.48 | 61.96 | 45.47 | 35.68 | 45.29 | 41.01 | 67.01 | 66.07 | 65.40 | 60.40 |
| TCP+LA | 22.03 | 14.08 | **66.85** | 61.64 | 49.20 | 38.48 | 48.89 | 43.12 | 67.28 | 66.40 | 66.14 | 60.96 |
| MMA+LA | 25.02 | 16.52 | 65.98 | 60.26 | 49.50 | 39.10 | 48.65 | 42.99 | 65.57 | 64.35 | 65.88 | 60.65 |
| Candle | 24.18 | 15.35 | 66.17 | 60.43 | 48.10 | 37.96 | 48.49 | 42.26 | 67.20 | 66.15 | 65.85 | 60.61 |
| LAMP | **26.24** | **17.83** | 66.73 | **62.10** | **50.47** | **40.62** | **50.40** | **44.64** | **68.97** | **67.64** | **67.44** | **62.46** |

Table 4: **Results of Long-tailed domain generalization benchmark** averaged over 5 runs. All methods are trained on an imbalanced ImageNet with `imb = 100` and directly evaluated on other four target datasets. We report the accuracy and macro-F1. The best results are marked in bold.

| Methods | ImageNet (Source) | | ImageNet-A | | ImageNet-V2 | | ImageNet-S | | ImageNet-R | |
|---|---|---|---|---|---|---|---|---|---|---|
| | Acc. | Mac. | Acc. | Mac. | Acc. | Mac. | Acc. | Mac. | Acc. | Mac. |
| CoOp+LA | 70.75 | 63.92 | 48.55 | 45.94 | 63.53 | 60.12 | 47.24 | 45.01 | 73.67 | 64.05 |
| CoCoOp+LA | 71.24 | 64.40 | 48.87 | 46.00 | 63.62 | 60.44 | 47.89 | 45.83 | 74.45 | 64.94 |
| MaPLe+LA | 70.18 | 63.35 | 49.32 | 45.32 | 63.70 | 60.45 | 48.05 | 45.69 | 74.68 | 65.02 |
| TCP+LA | 71.03 | 64.34 | 48.79 | 45.91 | 63.24 | 59.75 | 48.00 | 45.68 | 74.93 | 65.31 |
| MMA+LA | 70.65 | 63.98 | 50.12 | 46.12 | 63.55 | 59.97 | 47.65 | 45.32 | 75.34 | 65.96 |
| Candle | 70.92 | 64.62 | 49.16 | 45.28 | 62.70 | 59.46 | 48.38 | 45.94 | 75.20 | 65.80 |
| LAMP | **72.70** | **66.28** | **52.92** | **48.30** | **65.51** | **61.67** | **49.20** | **48.79** | **76.46** | **67.47** |

▶ **Results of few-shot benchmark.** To further show the robustness of LAMP and also align with most previous methods, we report the results of few-shot (16-shot) benchmarks in Table 2. We find that under the few-shot setting, LAMP still demonstrates superior performance: although it performs suboptimally in several datasets like Caltech101, DTD and EuroSAT, its averaged accuracy on base and new classes surpass the previous state-of-the-art by approximately 1.10% and 0.56% respectively. This further validates the strong robustness of LAMP in extreme scenarios.

▶ **Results of long-tailed cross-data transfer benchmark.** The results are shown in Table 3. We observe that LAMP largely outperforms previous methods by an average of 1.64%/1.50% in accuracy/macro-F1. Also, for challenging uncommon datasets like DTD and EuroSAT, LAMP yields considerable improvements, demonstrating its strong generalization ability across different datasets.

▶ **Results of long-tailed domain generalization benchmark.** Results are shown in Table 4. We observe that: first, LAMP demonstrates the best performance during the source training phase, surpassing the previous state-of-the-art

Table 5: Results of efficiency comparisons.

| Method | Time / epoch | Tun. Params | HM-A | HM-M |
|---|---|---|---|---|
| MaPLe | 1.47min | 3555K | 78.56 | 73.31 |
| MMA | 0.90min | 674K | 78.95 | 73.50 |
| LAMP | 1.08min | 470K | **80.93** | **75.66** |

by approximately 1.46%/1.94% in accuracy/macro-F1. This indicates that LAMP is relatively more adept at adapting to long-tailed source sets. Furthermore, LAMP also shows superiority when evaluated on four components of ImageNet, outperforming existing methods by an average of 1.50%/2.45% in accuracy/macro-F1 and demonstrating strong generalization ability to different domains.

▶ **Efficiency Study.** We further compare the complexity of LAMP and previous state-of-the-arts of the same magnitude in Table 5, in which we report the inference time of an epoch on EuroSAT, tunable parameters, harmonic mean of the accuracy (HM-A) and macro-F1 (HM-M) of base and new classes. The main computational overhead of LAMP lies in optimizing prompts

Table 6: Contribution of each component.

| component | ImageNet | | Food101 | |
|---|---|---|---|---|
| | HM-A | HM-M | HM-A | HM-M |
| CLIP | 68.05 | 62.45 | 89.25 | 88.83 |
| MPP | 70.05 | 64.08 | 90.87 | 89.50 |
| MSP | 70.94 | 64.64 | 91.21 | 89.54 |
| MPP+MSP | 71.23 | 64.70 | 91.10 | 89.87 |
| MPP+LBO | 72.42 | 65.83 | 92.06 | 90.84 |
| MPP+MSP+LBO | 72.96 | 66.67 | 92.45 | 91.15 |

and projectors, which is similar to previous state-of-the-arts. However, LAMP requires fewer tunable parameters, because the dimension of prompts is smaller than that of visual and textual hidden embeddings, as shown in the ablation studies. In addition to these efficiency gains, LAMP yields remarkable performance gains, validating its superiority.

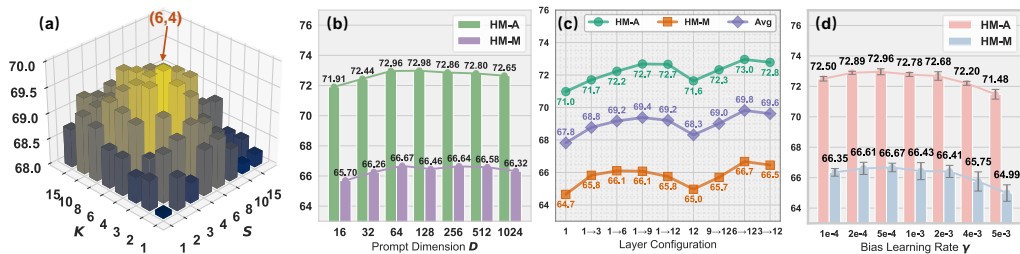

Figure 3: **The ablation studies** of (a) $K$ and $S$; (b) Prompt dimension $D$; (c) layer configuration; (d) bias (initial) learning rate, with standard variations. All ablation studies on conducted on ImageNet.

## 4.3 Ablation Studies

▶ **Different components.** Recall that LAMP comprises three essential components: Multimodal Prompt Pool (MPP), Modality-Shared Prompts (MSP), Load Balancing Optimization (LBO). We discover the effects of each component or their combination in Table 6 under the long-tailed B2N benchmark with ImageNet and Food101. We observe that using only MPP and MSP yields an improvement of approximately 2.5% in accuracy/macro-F1 over the baseline CLIP, with MSP yielding slightly more improvements than MPP due to its ability to capture cross-modal synergistic semantics. Furthermore, applying LBO to MPP leads to significant gains in accuracy and macro-F1, demonstrating that our load-balancing mechanism effectively promotes rebalancing. Finally, MPP+MSP+LBO achieves the best performance, indicating that three components work synergistically.

▶ **The length of shared prompts in MPP $K$ and extra prompts $S$.** We report the grid-search results (average of HM-A and HM-M) of these two hyper-parameters in Figure 3.a on long-tailed B2N benchmark. We find that excessively low values lead to more severe performance degradation; similarly, when the choices of K and S grows too large ($\geq 8$), we also observe a slight performance drop, possibly due to overfitting of training set. We find that $K = 6$ $J = 4$ are the optimal choices.

▶ **Dimension of prompts.** The dimension of prompts determines the upper bound of fine-tuning and also the size of the trainable parameters. We report the ablation studies in Figure 3.b on long-tailed B2N benchmark. Although both metrics increase significantly initially from 16, they plateau after 64 and even exhibit minor fluctuations. Meanwhile, higher choices entail surging computational costs. Balancing performance and efficiency, we select 64 as the optimal dimension.

▶ **Layers for adding prompts.** We ablate this factor in Figure 3.c on long-tailed B2N benchmark. We found that fine-tuning the first few layers ($\leq 9$) yields a performance gap of approximately 0.3% / 0.4% compared to fine-tuning the later layers ($\geq 6$), possibly because prompts can be more easily optimized within the high-level semantics represented by the deeper layers. We find 6-12 achieves optimal results.

Table 7: Results of different $\beta$.

| $\beta$ | HM-A | HM-M | Avg. |
|---|---|---|---|
| 0.00 | 71.95 | 65.70 | 68.83 |
| 0.20 | 71.49 | 66.02 | 69.25 |
| 0.50 | 72.67 | 66.50 | 69.59 |
| 1.00 | 72.79 | 66.58 | 69.69 |
| 2.00 | 72.96 | 66.67 | 69.82 |
| 4.00 | 72.89 | 66.04 | 69.47 |
| 8.00 | 72.84 | 65.96 | 69.40 |

▶ **The (initial) bias learning rate $\gamma$.** As aforementioned, we employ a cosine scheduler to adjust $\gamma$, and only conduct ablation studies regarding its initial value. Results are in Figure 3.d. We observe a severe performance drop when $\gamma$ employs a initial value over $1e-3$, accompanied by considerate variations in HM-M. This suggests that larger $\gamma$ values tend to cause instability in our LBO. Our experiments indicate that setting $\gamma = 5e-4$ generally achieves optimal results.

▶ **The balance coefficient $\beta$.** We report the results in Figure 7. We find that omitting the similarity loss ($\beta = 0$) resulted in an average performance degradation of approximately 1%. We argue that employing this loss facilitates further rebalancing and reduces fluctuations of the LBL during the initial training stages. $\beta = 2$ is the optimal. Ablation studies of $M$ is in **Appx. D**.

## 5 Conclusion

While most existing multimodal prompt learning methods are designed for the balanced dataset, real-world datasets always follow a long-tailed distribution, stressing the importance of designing prompting methods over imbalanced data. LAMP introduces three crafted and co-designed mechanisms: multimodal prompt pool, modality-shared prompts and load balancing optimization. Extensive comparisons and ablation studies validate the superiority and effectiveness of LAMP.

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

# Appendix
# LAMP: Long-tailed Multimodal Prompt Learning

The content of the **Appendix** is summarized as follows:

- in Sec. A, we conduct case study.
- in Sec. B, we introduce datasets and tasks.
- in Sec. C, we introduce methods for comparison.
- in Sec. D, we introduce extra ablation studies.
- in Sec. E, we give more experiments.
- in Sec. F, we provide statement of usage of LLM.
- in Sec. G, we provide reproducibility statement.

## A  CASE STUDY

To facilitate the understanding of strong theoretical expressivity of LAMP, in this section we aim to list examples which vanilla instance-level prompt learning fails but able to solve with LAMP. We first formulate our task as the following:

Refering  (Wang et al., 2023; Hu et al., 2024), we pick up the following case for discussion:

**Case A.1** *toy model: a pretrained single-layer single-head transformer $T$ consisting of an Attention layer $Attn$ and a MLP layer $MLP$:*

$$MLP(x) = \begin{cases} (W_2 W_1 + I)x, & [W_1 x]_0 > 0 \\ x, & [W_1 x]_0 \le 0 \end{cases} \tag{12}$$

*toy dataset: $Toy = \{(X_1 = [x_1, x_0], Y_1 = [y_1, y_0]), (X_2 = [x_2, x_0], Y_2 = [y_{21}, y_{20}])\}$, where $Y_1 \in C^{base}$ and $Y_2 \in C^{new}$.*

**Theorem A.2** *For vanilla prompt learning, there exists no $P \in R^{n_p \times d}$ that simultaneously satisfies $Y_0 = T([P, X_0])$ and $Y_1 = T([P, X_1])$; However, for LAMP, this is conditionally satisfiable.*

*Proof.* Some of this proof follows (Wang et al., 2023) but expands their conclusions. Generally, we first define the convex cone of any vector combination $\{\mathbf{v}_0, \mathbf{v}_1...\mathbf{v}_n\}$ as $Cone(\mathbf{v}_0, \mathbf{v}_1...\mathbf{v}_n) = \{x : \sum_{i=1}^{m} v_i \mathbf{v}_i, v_i > 0\}$. Considering that $W_2 W_1$ is a non-singular matrix, then we can find at most two points for the inverse projection for $MLP(x_0)$ from $\mathcal{Y}$ to $\mathcal{X}$, i.e $MLP^{-1}(y) = \{y, (W_2 W_1 + I)^{-1} y\}$. We rename them as $y_{10}$ and $y_{20}$.

Then, we can find $\mathbf{c}_1$ and $\mathbf{c}_2$, satisfying $\mathbf{c}_1, \mathbf{c}_2 \perp y_{10} - x_0$, $\mathbf{c}_1, \mathbf{c}_2 \perp y_{20} - x_0$, $(W_2 W_1 + I)^{-1} y_{10} - x_0$, $(W_2 W_1 + I)^{-1} y_{20} - x_0$, and $c_1 \perp c_2$. We can pick up $x_1$ and $x_2$ such that $Cone(-Attn(x_0, X_1), a - x_0) \cap Cone(-Attn(x_0, X_2), b - x_0) = \emptyset$, subject to $\forall a \in \{y_{10}, (W_2 W_1 + I)^{-1} y)_{10}\}$ and $\forall b \in \{y_{10}, (W_2 W_1 + I)^{-1} y)_{20}\}$.

Under the vanilla prompt learning scheme, there exists an $P$ in vanilla prompt learning that satisfies both $Y_0 = T([P, X_0])$ and $Y_1 = T([P, X_1])$. We also have:

$$Attn(x_0, [P, X_1]]) = \lambda(X_1, x_0, [P, X_1]) Attn(x_0, X_1) + \lambda(P, x_0, [P, X_1]) Attn(x_0, P)$$
$$Attn(x_0, [P, X_2]]) = \lambda(X_1, x_0, [P, X_2]) Attn(x_0, X_2) + \lambda(P, x_0, [P, X_2]) Attn(x_0, P) \tag{13}$$

the following reformulation can be made:

$$Attn(q_i, [P, X_0]) = \frac{q_i \sum_{j \in I_P} ([P, X_0]_j^T)^2}{q_i \sum_j [P, X_0]_j} + \frac{q_i \sum_{j \in I_{X_0}} ([P, X_0]_j^T)^2}{q_i \sum_j [P, X_0]_j}$$

$$= \frac{q_i \sum_{j \in I_P} [P, X_0]_j}{q_i \sum_j [P, X_0]_j} \frac{q_i \sum_{j \in I_P} ([P, X_0]_j^T)^2}{q_i \sum_{j \in I_P} [P, X_0]_j} + \frac{q_i \sum_{j \in I_{X_0}} [P, X_0]_j}{q_i \sum_j [P, X_0]_j} \frac{q_i \sum_{j \in I_{X_0}} ([P, X_0]_j^T)^2}{q_i \sum_{j \in I_{X_0}} [P, X_0]_j} \tag{14}$$

$$= \alpha \cdot Attn(q_0, P) + (1 - \alpha) \cdot Attn(q_0, X_0)$$

Table 8: Prompts used for all the datasets.

| Datasets | Prompts |
|---|---|
| ImageNet | "a photo of a [CLASS]." |
| Caltech101 | "a photo of a [CLASS]." |
| OxfordPets | "a photo of a [CLASS], a type of pet." |
| StanfordCars | "a photo of a [CLASS]." |
| Flowers102 | "a photo of a [CLASS], a type of flower." |
| Food101 | "a photo of a [CLASS], a type of food." |
| FGVCAircraft | "a photo of a [CLASS],a type of aircraft." |
| SUN397 | "a photo of a [CLASS]." |
| DTD | "[CLASS] texture." |
| EuroSAT | "a centered satellite photo of [CLASS]." |
| UCF101 | "a photo of a person doing [CLASS]." |
| ImageNet | "a photo of a [CLASS]." |
| ImageNet-V2 | "a photo of a [CLASS]." |
| ImageNet-S(Sketch) | "a photo of a [CLASS]." |
| ImageNet-A | "a photo of a [CLASS]." |
| ImageNet-R | "a photo of a [CLASS]." |

$I_P$ and $I_{X_0}$ represents their index set respectively. $\alpha$ is the coefficient controlling the balance between two types of attention. (Xu et al., 2023) provides in-depth discussions to this term, and finds that prompt tuning tends to learn less similar prompts for adaptation, but under the risk of discarding the pre-trained knowledge. The following also holds true:

$$Attn(x_0, [P, X_1]) + x_0 \in MLP^{-1}(y_{10}), \quad Attn(x_0, [P, X_2]) + x_0 \in MLP^{-1}(y_{20}) \quad (15)$$

This implies that, $Attn(x_0, P)$ must be in both $Cone(a - x_0, -Att(x_0, X_1))$ and $Cone(b - x_0, -Att(x_0, X_2))$ where $a \in \{y_{10}, (W_2 W_1 + I)^{-1} y_{10}\}$ and $b \in \{y_{10}, (W_2 W_1 + I)^{-1} y)_{20}\}$. This obey the aforementioned hypothesis, saying that the cones are distinct, meaning that prompt learning fails on learning this toy case.

We then show how LAMP do this. Starting from 14, we have:

$$A(x_0, [P_{y_0}^*, X_1]]) = \lambda(X_1, x_0, [P_{y_0}^*, X_1]) A(x_0, X_1) + \lambda(P_{y_0}^*, x_0, [P_{y_0}^*, X_1]) A(x_0, P_{y_0}^*)$$
$$A(x_0, [P_{y_1}^*, X_2]]) = \lambda(X_1, x_0, [P_{y_1}^*, X_2]) A(x_0, X_2) + \lambda(P_{y_1}^*, x_0, [P_{y_1}^*, X_2]) A(x_0, P_{y_1}^*) \quad (16)$$

Where $Attn$ is abbreviated as $A$, $P_{y_c}^*$ stands for the collection of $\{P_i\}, \{P_i^v\}, \{P_{i,c}^t\}$. Similarly, We can derive:

$$Attn(x_0, [P_{y_0}^*, X_1]) + x_0 \in MLP^{-1}(y_{10}), \quad Attn(x_0, [P_{y_1}^*, X_2]) + x_0 \in MLP^{-1}(y_{20}) \quad (17)$$

We ask $Attn(x_0, P_{y_0}^*)$ to in both $Cone(a - x_0, -Att(x_0, X_1))$, while $Attn(x_0, P_{y_1}^*)$ to in $Cone(b - x_0, -Att(x_0, X_2))$. By fixing the shared prompts and only tune the extra visual or textual prompts, we can find corresponding optimal $P_i^{v+\dagger}$ and $P_i^{t+\dagger}$ such at the above requirements can be satisfied. However, if the extra prompts are discarded from LAMP, the theoretically expressivity upperbound for LAMP may be the same as vanilla prompt methods.

## B   DETAILS OF DATASETS AND TASKS

Detailed introduction of datasets are listed here:

- DTD (Describable Texture Dataset) is a texture datasets that containing 5,640 images with the size ranging from 300x300 to 640x640. The dataset has 47 classes.

- Food-101 contains 101,000 images concerning different types of food, with a uniformed size of 512x512. These images are from 101 classes, where each class consists of 750 images for training and 250 for testing.

- UCF-101 is a action recognition dataset containing 13,320 videos from 101 action classes. The total length of videos is 27h. For multimodal learning, these videos are sampled into 11422 images, with 7639 for training and 3783 for testing.

- SUN397 is a scene understanding dataset containing 108,754 images from 397 scene classes. Each class has as least 100 images.

- EuroSAT is a land-use 10-class classification dataset based on Sentinel-2 satellite images. It has 27,000 images with the size of 64x64. Each class has about 2000-3000 images.

- FGVC-Aircraft is a aircraft dataset containing 10200 hih-resolution images, including 102 classes with each contains 100 images.

- StanfordCars is a fine-grained car classification dataset, containing 16185 images from 196 classes. The training set contains 196 classes, and the number of training and tests 8144 and 8041.

- Caltech-101 is a classical object recognition dataset, containing 101 classes and 9000 images. The classes are imbalancedly distributed.

- OxfordFlowers contains 102 classes of flowers that commonly seen in British, with 40-258 classes in each class.

- OxfordPets contains 7349 images from 37 classes, with each image annotated with the segmentation and class label.

- ImageNet(1K) is a subset of ImageNet full dataset and contains 1000 classes. There're a total of 1,281,167 training images, 50,000 validation images and 100,000 testing images.

- ImageNet-V2 is a extended version of Imagenet, aiming to deal with the overfitting issue of original experiments. It includes three testsets: "Threshold0.7", "MatchedFrequency" and "TopImages", which are sampled from different strategies.

- ImageNet-R (ImageNet-Rendition) is a special testset for renditions. It includes renditions of 200 ImageNet classes, resulting in 30,000 image.

- ImgeNet-A (ImageNet-Adversarial) is proposed for discovering the model's robustness to adversarial attacks. These modifications for attack are nearly invisible to human but would result in mis-classification.

- ImageNet-S (ImageNet-Sketch) is a testset for sketch arts. It concludes sketch images for the same classes as original ImageNet.

We also list the prompts used for these datasets in Table 8.

## C  DETAILS OF METHODS FOR COMPARISON

- CoOp: CoOp (Zhou et al., 2022b) firstly introduces the prompt engineering in Natural Language Processing to Multimodal models. It adds extra prompts to the original input text to learn the optimal textual templates for downstream datasets.

- CoCoOp: CoCoOp (Zhou et al., 2022a) improves CoOp by constructing a vision-conditional textual prompt. The outputs of vision encoder are used to construct the textual prompts, thus achieving multi-modal interactions.

- MaPLe: MaPLe (Khattak et al., 2023a) introduces a layer-wise prompt learning technique, where each layer is plugged in a textual prompt, and the textual prompt is also used to generate a layer-wise visual prompt.

- TCP: TCP (Yao et al., 2024) proposes to incorporate final class embeddings, to further enhance the discriminability of prompts. Specifically, it introduces a Textual Knowledge Embedding (TKE) module to transfer the final class embeddings to the same space of intermediate textual tokens.

- MMA: MMA (Yang et al., 2024) introduces a modality-part-shared adapter between two encoders. Concretely, intermediate visual and textual tokens are firstly projected to a joint space, then processed with a modality shared MLP, and transforms to their original space.

- (Extra) LA: Logit Adjustment Loss (LA) (Ren et al., 2020) is a tailored cross-entropy loss for imbalanced learning, where each term in softmax function are appened with an extra term denoting the class cardinality of the corresponding class.

## D  FURTHER ABLATION STUDIES

Table 9: Results of different prompt tuning methods over 10 datasets under the imbalance ratios of 100. Following (Shi et al., 2024a), the models are trained over an imbalanced training set and inference over both base and new classes; we report the harmonic mean of testing accuracy over base and new classes. The best results are marked with **bold**.

|  | Cal. | Ox. | SC. | FLw. | Food. | FA. | SUN. | DTD. | ES. | UDF. | Avg |
|---|---|---|---|---|---|---|---|---|---|---|---|
| MaPLe+LA | 95.30 | 93.57 | 70.65 | 84.25 | 88.47 | 34.08 | 76.78 | 63.27 | 78.1 | 78.56 | 76.30 |
| TCP+LA | 94.47 | 92.90 | 71.76 | 85.4 | 89.67 | 32.70 | 77.00 | 65.49 | 80.3 | 79.03 | 76.87 |
| MMA+LA | **96.97** | **93.94** | **73.79** | 84.74 | 88.98 | 33.26 | **76.69** | 66.08 | 83.30 | 79.07 | 77.68 |
| Candle | 95.09 | 93.71 | 72.60 | 83.27 | 88.06 | 37.90 | 75.34 | 65.76 | 80.06 | 81.59 | 77.33 |
| LAMP | 95.75 | 93.00 | 73.13 | **85.36** | **90.21** | **38.47** | 76.22 | **66.70** | **83.54** | **81.98** | **78.43** |

Table 10: Results of different prompt tuning methods over 10 datasets under the imbalance ratios of 200. Following (Shi et al., 2024a), the models are trained over an imbalanced training set and inference over both base and new classes; we report the harmonic mean of testing accuracy over base and new classes.

|  | Cal. | Ox. | SC. | FLw. | Food. | FA. | SUN. | DTD. | ES. | UDF. | Avg |
|---|---|---|---|---|---|---|---|---|---|---|---|
| MaPLe+LA | 92.82 | 94.29 | 69.73 | 82.06 | 88 | 31.29 | 75.84 | 60.47 | 76.76 | 75.82 | 74.70 |
| TCP+LA | **94.01** | 93.05 | 70.46 | **84.77** | **89.84** | 32.64 | 76.5 | 62.56 | 79.74 | 78.48 | 76.20 |
| MMA+LA | 90.56 | 94.81 | 70.09 | 81.35 | 88.4 | 29.08 | 77.13 | 60.04 | 82.09 | 77.74 | 75.12 |
| Candle | 92.63 | 92.89 | 71.4 | 82.86 | 86.01 | 35.29 | 75.08 | **63.34** | 79.17 | 79.3 | 75.79 |
| LAMP | 93.08 | **94.87** | **71.98** | 82.35 | 89.26 | **37.3** | 75.87 | 63.34 | 79.85 | 80.35 | **76.82** |

**The impact of $M$.** We further discuss $M$'s impacts in this part. By fixing $K$ as 6, we follow (Dong et al., 2022a) to alter $M$ from 10 to 30. Results conducted on ImageNet with Long-tailed B2N benchmark are shown in the Figure 11. We find that the performance gain by increasing $\beta$ over 20 is quite minor (0.04%), which may result from that $M$ does not effectively influence the actual number of activated prompts; As an trade-off between performance and efficiency, we set $M$ to 20.

Table 11: Results of $M$.

| $M$ | HM-A | HM-M |
|---|---|---|
| 10 | 72.28 | 65.96 |
| 12 | 72.40 | 66.12 |
| 15 | 72.87 | 66.50 |
| 20 | 72.96 | 66.67 |
| 25 | 73.00 | 66.65 |
| 30 | 72.96 | 66.48 |

# E  MORE EXPERIMENT RESULTS

We provide more experiments under more imbalanced settings here, including base-to-new experiments under imbalance factor of 100/200, and 1/2/4/8-shot base-to-new experiments. Results for base-to-new with imbalance factors of 100/200 are jointly reported in Table 9/10. Results for base-to-new with 1/2/4/8-shot are reported in Table 12 / 13 / 14 / 15.

For 100/200 imbalance factor base-to-new experiments, we can find that LAMP also yields the best accuracy in these settings. While most existing methods suffers from dramatic performance degradation under these severe settings, and some even performs worse than zero-shot CLIP under several datasets, LAMP shows its unqiue robustness, outperforming the second-best results by and average of 0.71% under imbalance ratio 100 and 0.52% under imbalance ratio 200. These highlights that LAMP's design is strong enough to handle extreme circumstances.

For 1/2/4/8-shot base-to-new experiments, we can find that: (1) LAMP consistently achieves the best accuracy in all of these settings, with an average boosting of 1.71%, 1.84%, 1.25%, 2.42% compared with previous state-of-the-arts. Also, LAMP performs quite well over integrated and sophisticated datasets like ImageNet, DTD and EuroSAT; on sparse and cardinality-limited dataset like FGVCAircraft, the performance of LAMP may be not always achieves superiority, which may due to the enlarged randomness. (2) With the increment of shots from 1-8, basically all methods show a sharp increment of accuracy, while that of LAMP is larger than others. (3) We find that under 1/2-shot settings, previous state-of-the-art MMA even performs worse than text-only prompt methods CoOp and CoCoOp. We suggest that this may result from the special design in MMA, which leverages a directly shared bottleneck between visual and textual tokens to achieve modality interaction. This technique is especially hard-to-optimize when faced with limited data.

Table 12: **Comparison between LAMP and other methods under 1-shot base-to-new settings.** We report the Harmonic Mean (HM) over base sets and new sets. The best and second-best results are marked with **bold** and underlined.

| Method | Average | ImageNet | Caltech101 | OxfordPets | StanfordCars | Flowers102 |
|--------|---------|----------|------------|------------|--------------|------------|
| CoOp | 67.56 | 66.33 | 92.60 | 90.37 | 67.43 | 77.53 |
| CoCoOp | 66.79 | 69.43 | 93.83 | 91.27 | 67.22 | 72.08 |
| MaPLe | 69.27 | 62.67 | 92.57 | 89.10 | 66.60 | 83.30 |
| MMA | 69.28 | 69.17 | **92.90** | 91.23 | 67.87 | 83.60 |
| LAMP | **70.99** | **70.69** | 91.81 | **93.64** | **68.64** | **84.39** |
| Method | Food101 | FGVCAircraft | SUN397 | DTD | EuroSAT | UCF101 |
| CoOp | 84.33 | 21.37 | 66.77 | 50.23 | 54.93 | 71.23 |
| CoCoOp | **85.65** | 12.68 | **68.33** | 48.54 | 55.33 | 70.30 |
| MaPLe | 80.50 | 26.73 | 64.77 | 52.13 | **71.80** | 71.83 |
| MMA | 83.03 | **28.73** | 64.00 | 52.27 | 55.07 | 74.17 |
| LAMP | 81.64 | 27.02 | 64.19 | **54.69** | 68.24 | **76.04** |

Table 13: **Comparison between LAMP and other methods under 2-shot base-to-new settings.** We report the Harmonic Mean (HM) over base sets and new sets. The best and second-best results are marked with **bold** and underlined.

| Method | Average | ImageNet | Caltech101 | OxfordPets | StanfordCars | Flowers102 |
|--------|---------|----------|------------|------------|--------------|------------|
| CoOp | 70.65 | 67.07 | 93.07 | 89.80 | 70.50 | 87.33 |
| CoCoOp | 67.65 | 69.78 | 94.82 | 92.64 | 68.37 | 75.79 |
| MaPLe | 72.58 | 65.10 | 93.97 | 90.87 | 71.60 | 88.93 |
| MMA | 72.08 | **70.37** | 94.00 | 91.97 | 71.77 | 90.30 |
| LAMP | **74.42** | 70.34 | **95.53** | **94.64** | **72.46** | **91.24** |
| Method | Food101 | FGVCAircraft | SUN397 | DTD | EuroSAT | UCF101 |
| CoOp | 84.40 | 26.20 | 66.53 | 53.60 | 65.17 | 73.43 |
| CoCoOp | **86.22** | 15.06 | **69.03** | 52.17 | 46.74 | 73.51 |
| MaPLe | 81.47 | 30.90 | 67.10 | 55.50 | **78.30** | 74.60 |
| MMA | 82.50 | **31.90** | 67.17 | 56.90 | 59.80 | 76.17 |
| LAMP | 84.09 | 28.81 | 68.14 | **62.99** | 72.24 | **78.16** |

# F   USAGE OF LLMS

LLMs are only for improving the writing of this paper.

# G   REPRODUCIBILITY

The code is available in Anonymous Github Link.

Table 14: **Comparison between LAMP and other methods under 4-shot base-to-new settings.** We report the Harmonic Mean (HM) over base sets and new sets. The best and second-best results are marked with **bold** and underlined.

| Method | Average | ImageNet | Caltech101 | OxfordPets | StanfordCars | Flowers102 |
|---|---|---|---|---|---|---|
| CoOp | 74.02 | 68.73 | 94.40 | 92.57 | 74.47 | 92.17 |
| CoCoOp | 71.21 | 70.39 | **94.98** | 92.81 | 69.39 | 78.40 |
| MaPLe | 75.37 | 67.70 | 94.43 | 91.90 | 75.30 | 92.67 |
| MMA | 76.38 | 71.00 | 94.33 | 92.33 | 76.50 | 93.00 |
| LAMP | **77.63** | **71.14** | 93.94 | **93.04** | **77.24** | **94.09** |
| Method | Food101 | FGVCAircraft | SUN397 | DTD | EuroSAT | UCF101 |
| CoOp | 84.47 | 30.83 | 69.97 | 58.70 | 70.80 | 77.10 |
| CoCoOp | 86.88 | 24.79 | 70.21 | 55.04 | 65.56 | 74.82 |
| MaPLe | 81.77 | 34.87 | 70.67 | 61.00 | **84.50** | 78.47 |
| MMA | 82.13 | **37.57** | 69.97 | 63.93 | 79.40 | 80.10 |
| LAMP | **82.94** | 36.36 | **72.14** | **68.83** | 83.54 | **80.69** |

Table 15: **Comparison between LAMP and other methods under 8-shot base-to-new settings.** We report the Harmonic Mean (HM) over base sets and new sets. The best and second-best results are marked with **bold** and underlined.

| Method | Average | ImageNet | Caltech101 | OxfordPets | StanfordCars | Flowers102 |
|---|---|---|---|---|---|---|
| CoOp | 76.98 | 70.63 | 94.37 | 91.27 | 79.30 | 94.97 |
| CoCoOp | 72.96 | 70.63 | 95.04 | 93.45 | 70.44 | 84.30 |
| MaPLe | 78.89 | 70.30 | 95.20 | 92.57 | 79.47 | 95.80 |
| MMA | 79.57 | 71.77 | 95.37 | 92.77 | 81.40 | **95.97** |
| LAMP | **81.99** | **72.83** | **97.26** | **93.67** | **93.77** | 94.2 |
| Method | Food101 | FGVCAircraft | SUN397 | DTD | EuroSAT | UCF101 |
| CoOp | 82.67 | 39.00 | 71.53 | 64.77 | 78.07 | 84.93 |
| CoCoOp | 86.97 | 26.61 | 70.84 | 58.89 | 68.21 | 73.32 |
| MaPLe | 83.60 | 42.00 | **73.23** | 66.50 | 87.73 | 92.33 |
| MMA | 83.00 | **44.83** | 72.30 | 67.97 | 86.47 | 92.37 |
| LAMP | **87.82** | 35.97 | 71.46 | **71.69** | **90.08** | **93.14** |

Table 16: Hyperparam setting for LAMP.

| Config field | Subfield | base-to-new imb | base-to-new few-shot | cross-dataset / domain-generalization |
|---|---|---|---|---|
| dataloader | train:batch-size | 4 | 32 | 32 |
| | test:batch-size | 100 | 100 | 100 |
| | num_workers | 8 | 8 | 8 |
| | size | [224,224] | [224,224] | [224,224] |
| input | interpolation | bicubic | bicubic | bicubic |
| | pixel_mean | | [0.4814,0.4578,0.4082] | |
| | pixel_std | | [0.2686, 0.2613, 0.2757] | |
| | transforms | | ["random_resized_crop", "random_flip", "normalize"] | |
| | name | adamw | adamw | adamw |
| optim | lr | 0.003 | 0.001 | 0.001 |
| | max_epoch | 50 | 50 | 1 |
| | lr_scheduler | cosine | cosine | cosine |
| | warmup_epoch | 1 | 1 | 1 |
| | warmup_type | constant | constant | constant |
| | warmup_cons_lr | 1e-5 | 1e-5 | 1e-5 |
| | print_freq | 5 | 5 | 5 |
| train | backbone.name | VIT-B/16 | VIT-B/16 | VIT-B/16 |
| model | share_dim | 32 | 32 | 32 |
| trainer.prolt | share_layers | | [3,4,5,6,7,8,9,10,11,12] | |
| | share_token_num | 2 | 4 | 4 |
| | extra_token_num | 2 | 4 | 4 |
| | prec | amp | amp | amp |