# OpenReview forum: "LAMP: Long-tailed Multimodal Prompt Learning for Vision-Language Models"
_ICLR.cc/2026/Conference — Submitted to ICLR 2026_

### Official Review · Reviewer_bA4V · 2025-10-26

**Soundness:** 2
**Presentation:** 2
**Contribution:** 2
**Rating:** 2
**Confidence:** 3

**Summary:**

This paper addresses prompt learning for long-tailed multimodal settings by introducing LAMP, composed of three modules—Multimodal Prompt Pool (MPP), Modality-Shared Prompts (MSP), and Load-Balancing Optimization (LBO). The work aims to achieve balanced adaptation of frozen Vision-Language Models under imbalanced distributions.

**Strengths:**

1. The idea of addressing long-tailed distribution issues in multimodal prompt learning is meaningful.

2. Experimental results validate the effectiveness of the proposed method compared with the baselines.

**Weaknesses:**

1. The paper lacks comparisons with more recent multimodal prompt learning methods such as MMRL, which share structural similarities (e.g., shared prompts and cross-modal interaction modules). The absence of these baselines makes it unclear whether the proposed improvements of LAMP stem from genuine novelty or merely incremental design differences.

2. The paper shows incremental novelty, combining existing prompt-tuning and load-balancing ideas without offering fundamentally new insights or mechanisms specific to long-tailed multimodal learning.

3. There is no visualization or quantitative measure showing improved balance in embedding space.

4. Some baselines (CoOp, CoCoOp, MaPLe) are not designed for long-tailed setups; simply adding Logit-Adjusted loss may not make them comparable.

5. The paper lacks qualitative analysis, as it provides no visualization of prompt activations, load-balancing dynamics, or cluster separability, making the claimed interpretability and feature re-clustering effects speculative.

6. Writing and presentation issues, such as inconsistent notation (e.g., “MMP” should be “MPP” in Section 3.2), unclear variable definitions (e.g., **t** seems to denote the textual prompt at line 202, **s** in Eq. 6 may refer to **vₛₕ**), and possible typos in formulas (e.g., (**vₘ, tₘ**) should be (**vₖ, tₖ**), make the paper confusing to understand.

7. This is not a weakness, but for the sake of rigor, it is recommended that the authors include introductions of Candle and LFA in Appendix C.

[1] MMRL: Multi-Modal Representation Learning for Vision-Language Models. Yuncheng Gu et al.,CVPR2025.

**Questions:**

The authors claim that LAMP alleviates imbalance through feature clustering, but there is no mathematical or statistical evidence demonstrating how clustering via prompts leads to more balanced class representations. Could you provide quantitative analyses or theoretical justification to support this claimed mechanism?

---

### Official Review · Reviewer_3tft · 2025-10-30

**Soundness:** 2
**Presentation:** 2
**Contribution:** 3
**Rating:** 4
**Confidence:** 4

**Summary:**

This paper proposes LAMP (Long-tailed multimodal prompt tuning), a prompt tuning framework to address the problem of adapting VLMSs to downstream tasks with long-tailed (imbalanced) data distributions. Moreover, LAMP introduces three co-designed mechanisms: a Multimodal Prompt Pool, Modality-Shared Prompts, and Load Balancing Optimization to boost feature clustering and mutual learning of head and tail classes.

**Strengths:**

1. It's the first study of prompt learning for VLMs under long-tailed multimodal dataset settings.

2. The experiments are conducted across 15 datasets under four different settings.

**Weaknesses:**

1.The LBO mechanism aims to enforce a uniform prompt selection frequency, which is in potential conflict with the MPP's goal of selecting prompts based on feature similarity. The paper does not analyze whether LBO, in its quest for balance, might override the feature-clustering ability of MPP, especially if one prompt is genuinely the best "expert" for a large head class.

2.LBO draws on MoE, and its goal is to force the activation frequency of all prompts (experts) to tend towards a uniform distribution. Why must the optimal prompt distribution of a long-tail data set be uniform?

3.The method's effectiveness is only demonstrated on CLIP ViT-B/16. It is unclear if these gains will generalize to other VLM families (e.g., BLIP, EVA-CLIP) or different backbone sizes (e.g., ViT-L).

4.The novelty of the proposed framework appears limited, as its core components are largely adaptations of well-established techniques. Specifically, the concept of MPP that uses feature similarity or MoE principles for prompt selection has already been extensively explored in prior work [1-4]. Furthermore, MSP is inspired by previous cross-modality prompts studies such as MaPLe, Mmrl. Same as LBO.

[1] Enhancing Adversarial Robustness of Vision Language Models via Adversarial Mixture Prompt Tuning

[2] One Prompt is not Enough: Automated Construction of a Mixture-of-Expert Prompts

[3] MoPD: Mixture-of-Prompts Distillation for Vision-Language Models

[4] Mixture of Prompt Learning for Vision Language Models

**Questions:**

1.The "Implementation Details" (Sec 4.1) state that $\beta=1$ and "All prompt dimension is set to 32, and we have a concrete discussion on this in the appendix". However, the "Ablation Studies" (Sec 4.3) conclude that the optimal choices are $\beta=2$ and a optimal dimension of 64. So which set of parameters was used for the main experiments? And there is no discussion in the appendix.

2.Line 356 states "The length and depth of shared prompts are set to 9 (3→12) and 2, respectively" Are the lengths and depths  written in reverse? Why is it set to 3-12?

3.There are typographical errors in key definitions. For instance, in line 202, the text prompt is defined as $v_K$ . Furthermore, Eq. (6) uses an undefined variable  $s$ .

4.The efficiency study ignores the inference latency introduced by the MPP.

---

### Official Review · Reviewer_aiwt · 2025-11-01

**Soundness:** 3
**Presentation:** 4
**Contribution:** 2
**Rating:** 2
**Confidence:** 4

**Summary:**

This paper introduces a prompt learning method for vision-language models (VLMs) in long-tailed scenarios. To address prediction bias toward head classes, the authors propose three techniques: Multimodal Prompt Pool (MPP), Modality-Shared Prompts (MSP), and Load Balancing Optimization (LBO). The method inserts learnable multimodal prompts (expert-specific and shared) into intermediate layers to enhance feature clustering. Experiments are conducted on standard prompt learning datasets adapted to long-tailed distributions.

**Strengths:**

1. The paper is easy-to-follow and the problem is novel and meaningful in prompt learning,
2. The main results demonstrate notable performance gains, and ablation studies comprehensively cover hyper-parameters (e.g., prompt dimensions, expert counts).

**Weaknesses:**

1. Methodological Justification: The core claim that MPP+MSP+LBO mitigates head-class bias, requires stronger validation.

- Show expert activation patterns for head vs. tail classes (e.g., attention maps, class-wise expert utilization rates). Ideally, different experts should specialize in distinct class subsets.

- Explain how mutual learning between head/tail classes is achieved and how overfitting to head classes is prevented (e.g., gradient analysis, feature similarity metrics).

2. Complexity & Sensitivity: The framework is over-parameterized, with hyper-parameters (e.g., MPP size, MSP dimensions, LBO weights) highly sensitive to performance.

3. Comparison Gaps: Include more baselines tailored for long-tailed recognition (e.g., class re-balancing, logit adjustment, decoupling).

**Questions:**

1. What is the value of M (the size of multimodal prompt pool)? How does M affect the results?
2. What are the results of directly pre-training comparison methods on long-tailed datasets without any long-tailed tricks?

---

### Meta-Review · Area_Chair_qr2W · 2025-12-02

**Summary:**

This paper introduces a prompt learning method for vision-language models in long-tailed scenarios. However, reviewers have multiple concerns about the methodology (e.g.,  the LBO mechanism, expert activation patterns, mutual learning), the novelty, the effectiveness, the experimental analysis and sensitivity of hyper-parameters. All reviewers give negative rating scores and the authors do not provide a rebuttal. Thus, I recommend to reject this manuscript.

**Reviewer Concerns:**

The authors do not provide rebuttal and all reviewers' concerns have not been addressed.

**Reviewer Scores:**

The reviewers may not change their rating scores.

---

### Decision · Program_Chairs · 2026-01-26

Reject